# *OCIAD2* as a novel prognostic and therapeutic biomarker for pancreatic cancer: A study based on transcriptomic signature and bioinformatics analysis

Zhongyuan Cui¹, Xia Lei¹, Yani Gou¹, Zhixian Wu²☯*, Xiaojun Huang¹☯*

**1** Department of Gastroenterology, The Second Hospital & Clinical Medical School, Lanzhou University, Lanzhou, China, **2** Department of Hepatobiliary Disease, 900th Hospital of the Joint Logistics Support Force (Dongfang Hospital of Xiamen University, School of Medicine, Xiamen University), Fuzhou, Fujian, China

☯ These authors contributed equally to this work.
* zxwu@xmu.edu.cn (ZW); huangxj@lzu.edu.cn (XH)

## Abstract

### Background

It is urgent to explore the potential biomarkers for pancreatic cancer (PC) prognosis and treatment to improve patients' outcomes.

### Methods

Firstly, we performed an integrated bioinformatics analysis based on extensive transcriptome data from 615 PC tumors and 329 adjacent tissues, screening for genes with prognostic value. We then validated the prognostic value of *OCIAD2*, *DCBLD2*, and *SAMD9* in different datasets and analyzed their expression levels in single-cell sequencing datasets of normal, paracancer, primary, and metastatic tissues. Next, we further explored the carcinogenic effect after knocking down the expression of *OCIAD2* in PC cancer cell line. Finally, a drug sensitivity analysis was conducted.

### Results

Differentially expressed genes (DEGs) analysis identified 22 DEGs: *ACSL5*, *ANTXR1*, *AP1S3*, *ATP2C2*, *B3GNT5*, *C15orf48*, *CAPG*, *CTSK*, *DAPP1*, *DCBLD2*, *GPX8*, *HEPH*, *IFI44*, *KRT23*, *NCF2*, *OCIAD2*, *SAMD9*, *SLC39A10*, *ST6GALNAC1*, *TBC1D2*, *TMSB10* and *TSPAN5* with prognostic value in PC, though the related function and mechanism are still unclear. Single-cell sequencing results indicated that *OCIAD2* was prominently expressed in ductal cells of primary and metastatic tumors. The expression levels of *OCIAD2* mRNA and protein were the highest in pancreatic tumor tissues. Mechanism studies revealed that *STAT1* and *STAT2* in the JAK-STAT pathway and *CCND1*, *CDK1,* and *CDK2* in the cell cycle pathway were significantly

**Data availability statement:** The public data used in this study are all available in the citations corresponding to the Methods section (or by entering the corresponding numbers in the referenced databases). The raw sequence data of BxPC-3 reported in this paper have been deposited in the Genome Sequence Archive in National Genomics Data Center, China National Center for Bioinformation / Beijing Institute of Genomics, Chinese Academy of Sciences (GSA-Human: HRA007928) that are publicly accessible at https://ngdc.cncb.ac.cn/gsa-human. The count and FPKM data provided in S3 Data.

**Funding:** This study was supported by the 2022 Drug Supervision Scientific Project of Gansu Provincial Food and Drug Administration (Grant No. 2022GSMPA0015, and 2022GSMPA0016 to XJH), Science and Technology Innovation Joint Fund Project of Fujian Province (Grant No. 2023Y9266 to ZXW), and the Cuiying Science and Technology Innovation Program of the Second Hospital of Lanzhou University (Grant No. CY2024-QN-B10 to YNG). The funders had no role in study design, data collection and analysis, decision to publish, or preparation of the manuscript.

**Competing interests:** The authors have declared that no competing interests exist.

down-regulated after *OCIAD2* knockdown. Drug sensitivity analysis identified 25 compounds significantly associated with *OCIAD2*.

### Conclusions

These results indicate that OCIAD2 is a potential prognostic biomarker and therapeutic target for PC patients.

### Author summary

Pancreatic cancer is the most malignant tumor, and there is no ideal targeted drug at present, so the prognosis of patients is very poor. There is an urgent need to find targets for evaluating prognosis and treatment. In this study, we identified a number of poorly understood but potentially important prognostic genes based on transcriptome data from a large number of pancreatic cancer samples. Also based on transcriptome data from pancreatic cancer samples and cell lines, we focused on the activation of JAK-STAT and cell cycle pathways by OCIAD2 overexpression in pancreatic cancer patients. Meanwhile, we also analyzed the sensitivity of patients with different OCIAD2 expression to 545 drugs and identified 25 important drugs. These results suggest that OCIAD2 is a potential novel biomarker for prognosis and targeted therapy in patients with pancreatic cancer, which deserves more attention and research.

### Introduction

Pancreatic cancer (PC) is one of the most malignant tumors. Recent tumor epidemiology shows its incidence ranks 10th and mortality ranks 6th [1]. Although PC does not have the highest incidence, PC has the worst 5-year survival rate. Recent studies report that the highest 5-year survival rate was only 12% [1,2]. To improve the PC prognosis, numerous clinical trials have been conducted, yet most have not met expectations. Reports showed that the failure rate of phase III clinical trials for PC was the highest among common solid cancers [3].

Biomarkers for diagnosis, treatment, and research indicate that clinical drug trials based on various mutations have not yielded satisfactory results [4–6]. Other novel targeted therapies have also failed to improve overall survival (OS) [7–9]. PC's ability to escape immune surveillance early in the disease haprognostic evaluation is are critical tool in PC precision medicine. Accurate biomarkers can better stratify patients and guide treatment plans. CA19–9 is currently the only biomarker used for PDAC, primarily for assessing recurrence and response to therapy [10]. Several currently reported biomarkers stem from small heterogeneous tumor samples, without extensive validation, posing a challenge to their reliability [11].

These reasons contribute to the difficulties in improving the 5-year survival for PC. Moreover, the incidence of PC is predicted to increase shortly, potentially surpassing

colorectal cancer and becoming the second leading cause of cancer-related death after lung cancer [12]. Despite progress in basic or translational research on PDAC biology, diagnosis, treatment, and prognosis in the past two decades, research lags behind other cancer types [13]. We conducted an integrated study with analysis of a large amount of published PDAC transcriptomic and clinical data while performing mechanistic experiments, aiming at identifying promising novel prognostic and therapeutic biomarkers for PC patients.

## Materials and methods

### Analysis of differentially expressed genes (DEGs) in pancreatic tumors and adjacent tissues

We used the keywords "pancreatic cancer", "pancreatic ductal adenocarcinoma", "pancreatic ductal carcinoma", or "Pancreatic adenocarcinoma (PAAD)" to search the Gene Expression Omnibus (GEO) database. We obtained the transcriptome data of PC and adjacent tissues from 7 independent studies, such as GSE102238 [14], GSE183795 [15], GSE71729 [16], GSE62452 [17], GSE28735 [18], GSE62165 [19], and GSE60979 [20] (Table A in S1 Text). For repeated probes or genes in these microarray expression data, we keep only the median value and then use the biomaRt (version 2.56.1) package [21] to filter protein-coding genes, keeping only those genes expressed in all tumor tissue samples for differential analysis. Differentially expressed genes were identified using the limma (version 3.56.2) package [22]. Fold change of ≥ 1.5 and adjusted p-values of ≤ 0.05 were used as a threshold for significant *DEGs*. After independent analysis of each group of data, the up-regulated and down-regulated DEGs were intersected respectively. Finally, Gene Ontology (GO) and Kyoto encyclopedia of genes and genomes (KEGG) enrichment analysis were performed using the clusterProfiler package (version 4.10.1) [23].

### Screening DEGs with prognostic value

The upregulated DEGs with unclear function and mechanism were searched through the PubMed database for subsequent analysis. First, we conducted a preliminary screening in the Kaplan-Meier Plotter database [24] to select genes with prognostic value. Next, the Cancer Genome Atlas (TCGA) pancreatic adenocarcinoma (PAAD) fpkm transcriptome and clinicopathological data were downloaded via UCSC Xena [25] to analyze the relationship between DEGs with prognostic value, prognosis, and other clinicopathological features. GSE79668 [26] dataset was used to investigate the relationship between DEGs and the prognosis of PC patients. Multivariate Cox analysis and visualization of survival using survival (version 3.5-8) (https://CRAN.R-project.org/package=survival) and survminer (version 0.4.9) Package (https://CRAN.R-project.org/package=survminer). The regplot (version 1.1) package (https://CRAN.R-project.org/package=regplot) was used to construct the nomogram based on the prediction model. The area under the time-dependent ROC curve of the prediction model was calculated using the timeROC package [27]. Decision curve analysis (DCA) by the ggDCA (version 1.2) package was used to evaluate the clinical net benefit of the prediction models (https://github.com/yikeshu0611/ggDCA).

### The mRNA expression levels of DCBLD2, OCIAD2, and SAMD9 in various tissues by single-cell sequencing

Single-cell sequencing datasets of normal tissues and PC were retrieved from the GEO database. Those are GSE155698 [28], GSE154778 [29], GSE197177 [30], GSE212966 [31], GSE229413 [32], and GSE156405 [33]. These datasets contained single-cell sequencing data of normal pancreas, adjacent normal, primary, and metastatic tumor tissues. The data was analyzed and visualized by R software (version 4.3.0, The R Foundation for Statistical Computing, Vienna, Austria) using Seurat (version 5.0.3) [34], harmony (version 1.2.0) [35], Dittoseq (Version 1.14.2) [36], and Scientomize (Version 2.1.2) packages. The analysis process is in S2 Text.

### The expression of DCBLD2, OCIAD2 and SAMD9 proteins in clinical specimens

The Human Protein Atlas (HPA) database [37] and Gene Expression Profiling Interactive Analysis (GEPIA) [38] were used to investigate the protein and mRNA expression levels of *DCBLD2*, *OCIAD2,* and *SAMD9* in normal and tumor pancreatic tissues, respectively.

## Cell culture

The BxPC-3 pancreatic cancer cell line was purchased from Suzhou Haixing Biosciens Co., Ltd. BxPC-3 cells were cultured in RPMI-1640 medium supplemented with 10% fetal bovine serum (FBS) (HyCyte, Cat# FBP-C520, China), 1% streptomycin and penicillin in an incubator at 37°C with 5% $CO_2$.

## siRNA transfection

The siRNA targeting *OCIAD2* was designed and synthesized by Beijing Tsingke Biotech Co., Ltd. The siRNA sequence was: Sense: GACUAGUCUACCAAGGUUA(dT)(dT), Anti-sense: UAACCUUGGUAGACUAGUC (dT)(dT). The negative control sequence was: Sense: UUCUCCGAACGUGUCACGUTT, Anti-sense: ACGUGACACGUUCGGAGAATT. TSnanofect V2 (Tsingke, Cat# TSV405, China) transfection reagent was purchased from Beijing Tsingke Biotech Co., Ltd. BxPC-3 cells were cultured in 6 or 24-well plates and transfected according to the TSnanofect V2 transfection reagent and siRNA instructions when the cells were in the logarithmic growth phase and used for subsequent experiments after 24 hours.

## RNA sequencing and bioinformatics analysis

To explore the potential mechanism *OCIAD2* promotes progress in PC, we conducted the transcriptome sequencing and analysis. siRNA targeting knockdown of *OCIAD2* and negative controls were transfected into BxPC-3 cells. After 48 hours, cells from each group were collected, and TRIzol reagent was added to extract total RNA. Transcript sequencing was performed by Tsingke Biotech (Beijing, China). RNA extraction and subsequent transcription sequencing library preparation followed the instructions provided by the manufacturer. The main Library preparation Kit used was VAHTS Universal V6 RNA-seq Library Prep kit for MGI (Cat# NRM604–01), and the sequencing platform was BGI DNBSEQ-T7 sequencer.

DEGs analysis was performed using the Limma (Version 3.56.2) package [22]. DEGs were defined as fold change of ≥ 1.2 and adjusted *p*-values of ≤0.05. The clusterProfiler (version 4.10.1) package [23] was used for GO, KEGG, and Gene Set Enrichment Analysis (GSEA). The inference of the pathways' status in each sample was conducted via the run_wmean and run_mlm functions of the decoupleR package (version 2.8.0) [39].

## Drug sensitivity analysis of potential targets

Based on the gene expression matrix of 38 pancreatic Cancer cell lines and 545 drug sensitivity data from the Cancer Therapeutics Response Portal (CTRP) [40], the oncoPredict package [41] was used to evaluate the 545 drug IC50 values of TCGA pancreatic cancer samples. The IC50 value is used as a measure of drug susceptibility, with a higher IC50 value indicating a lower sensitivity to the drug. Then, Pearson correlation coefficients were calculated between IC50 and *OCIAD2* expression for all drugs in the TCGA-PAAD cohort. A correlation coefficient of ≥0.6 and a p-value of ≤0.05 are considered to have a significant correlation. Finally, the top 10 compounds with the strongest positive and negative correlations were visualized.

## Statistical analysis

Statistical analysis and visualization were performed using R software (version 4.3.0, The R Foundation for Statistical Computing, Vienna, Austria). All experimental data were represented by the mean±SEM (Standard Error of the Mean). The comparison between multiple groups using analysis of variance (ANOVA) and unpaired two-tailed Student's t-test. Non-normal distribution data using nonparametric statistical analysis. $p < 0.05$ was considered statistically significant.

## Result

### The significant DEGs in the pancreatic tumor tissue

First, we performed differentially expressed genes analysis on the transcriptome data of pancreatic tumors and adjacent normal tissues from 7 independent studies, including GSE102238 [14], GSE183795 [15], GSE71729 [16], GSE62452 [17],

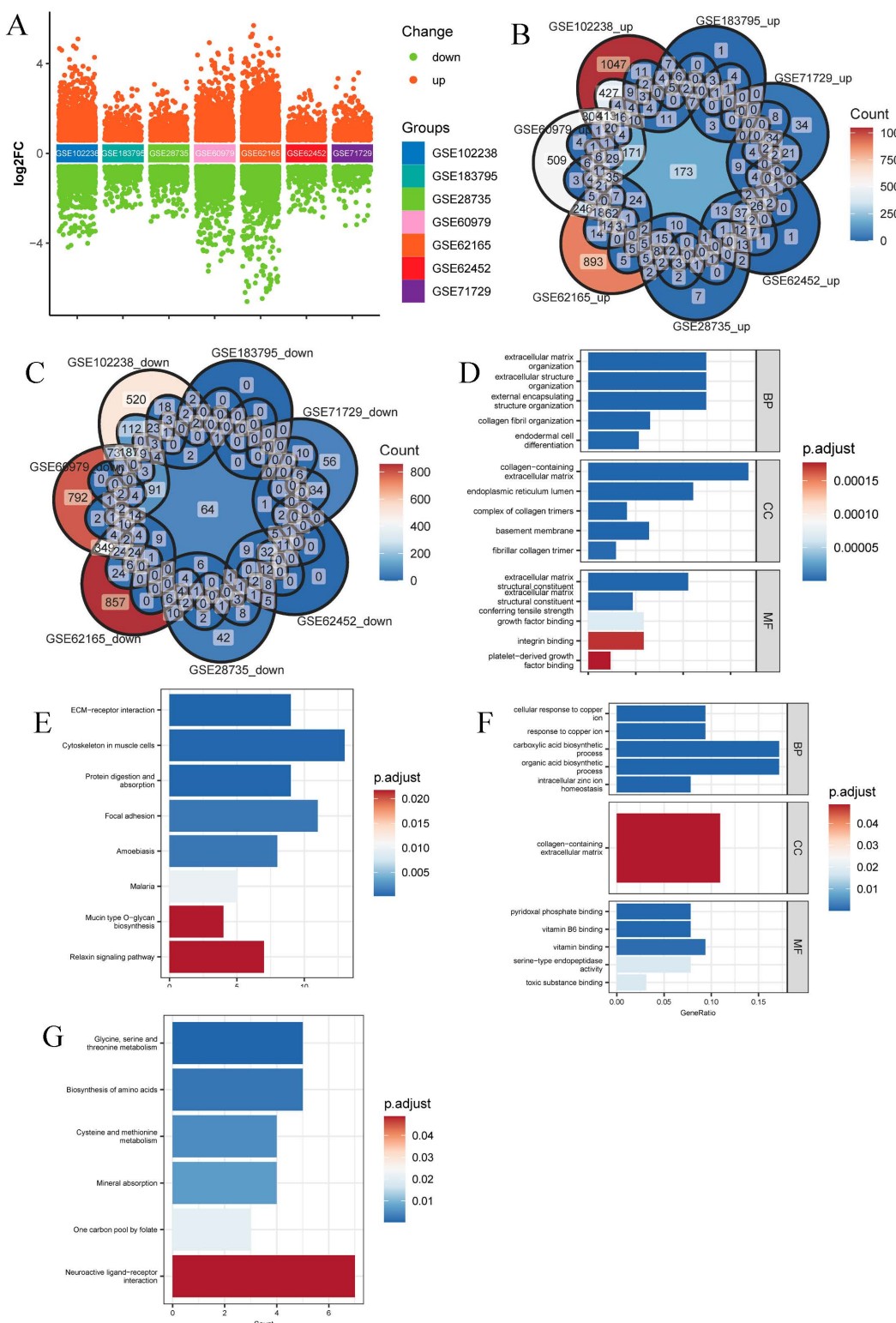

**Fig 1. Bulk sequencing data from pancreatic tumors and adjacent normal tissue based on 7 independent studies revealed significant DEGs that were prevalent in pancreatic cancer.** (A) Distribution of DEGs in each dataset. (B) The intersection of significantly up-regulated DEGs in each

data set, 173 genes were significantly up-regulated in all 7 datasets. (C) The intersection of DEGs was significantly down-regulated in each dataset, with 64 genes significantly down-regulated in all 7 datasets. (D-E) Enrichment results of significantly up-regulated DEGs in GO and KEGG analyses. (F-G) Enrichment results of significantly down-regulated DEGs in GO and KEGG analysis.

GSE28735 [18], GS E62165 [19] and GSE60979 [20] (Fig 1A). The results showed that the significantly up-regulated and down-regulated genes in each dataset: 3053 and 1291 for GSE102238, 619 and 271 for GSE183795, 499 and 312 for GSE71729, 689 and 302 for GSE62452, 754 and 448 for GSE28735, 2766 and 1919 for GSE62165, GSE60979 for 2263 and 1735 (Table B in S1 Text and S1 Data). Then, the intersection of DEGs was taken separately, and it was found that 173 genes were significantly up-regulated in all seven datasets (Fig 1B), and 64 genes were significantly down-regulated in all seven datasets (Fig 1C). Finally, GO and KEGG enrichment analyses were performed for up-regulated and down-regulated DEGs, respectively. The results showed that 173 up-regulated DEGs were mainly enriched in extracellular in biological process (BP), cellular component (CC), and molecular function (MF), respectively, extracellular matrix organization, collagen−containing extracellular matrix and extracellular matrix structural constituent (Fig 1D). KEGG enrichment analysis showed that up-regulated DEGs were mainly enriched in ECM−receptor interaction, cytoskeleton in muscle cells, protein digestion and absorption, and focal adhesion (Fig 1E). The 64 down-regulated DEGs in BP, CC, and MF were mainly enriched by cellular response to copper ion, collagen-containing extracellular matrix, and pyridoxal phosphate binding, respectively (Fig 1F). KEGG enrichment analysis showed that 64 down-regulated DEGs were mainly enriched in glycine, serine, and threonine metabolism, biosynthesis of amino acids, and cysteine and methionine metabolism (Fig 1G).

### DCBLD2, OCIAD2, and SAMD9 identified as novel biomarkers with independent prognostic value

We further investigated the 173 up-regulated DEGs by searching PubMed and Kaplan-Meier Plotter databases. DEGs with unclear function in PC and prognostic value were selected for subsequent studies. The results showed that 22 DEGs, including *ACSL5*, *ANTXR1*, *AP1S3*, *ATP2C2*, *B3GNT5*, *C15orf48*, *CAPG*, *CTSK*, *DAPP1*, *DCBLD2*, *GPX8*, *HEPH*, *IFI44*, *KRT23*, *NCF2*, *OCIAD2*, *SAMD9*, *SLC39A10*, *ST6GALNAC1*, *TBC1D2*, *TMSB10* and *TSPAN5*, possessed prognostic value and their related functions and mechanisms in PC had not been reported (Table 1). Next, we explored the relationship between these 22 DEGs and various clinicopathological parameters using the TCGA PAAD dataset (Fig 2A). Furthermore, multivariate Cox and stepwise regression analysis showed that *DCBLD2*, *OCIAD2*, *SAMD9*, age, and lymph node metastasis (LNM) status were independent prognostic factors for PC patients ($p < 0.05$) (Fig 2B). We constructed a Nomogram to predict the 1,3, and 5-year survival rates of patients based on the Cox model (Figs 2C and S1). Patients with a high predictive score in the prognostic model had a worse prognosis ($p < 0.0001$) (Fig 2D). The time-dependent receiver operating characteristic curve (ROC) and the area under the curve (AUC) illustrated the predictive sensitivity and specificity of this nomogram at 1, 3, and 5-year survival rates. The results of the analysis showed that the AUC for 1,3, and 5-year survival rates were 0.7, 0.77, and 0.68, respectively (Fig 2E). Clinical decision curve analysis results show that the prognostic model aided clinical decision-making to benefit patients (Fig 2F).

We validated the performance of the prognostic model in the GSE79668 [26] dataset. The results showed that the prognosis of PC patients with high prognostic model scores was significantly worse than those with low scores (Fig 2G). In the GSE79668 [26] dataset, the AUC of 1-, 3-, and 5-year survival rates of the prognostic model were 0.8, 0.86, and 0.96, respectively (Fig 2H). Similarly, the results of clinical decision curve analysis showed that the prognostic model aided clinical decision-making (Fig 2I).

### The expression levels of DCBLD2, OCIAD2, and SAMD9 in normal pancreatic, adjacent normal, primary, and metastatic tumor tissues

We retrieved 6 independent single-cell sequencing datasets, GSE155698 [28], GSE154778 [29], GSE197177 [30], GSE212966 [31], GSE229413 [32], and GSE156405 [33] from the GEO database. The datasets contained single-cell

**Table 1. Correlation between the DEGs mRNA expression and the survival of PAAD patients.**

| Gene | Survival | Cases | HR | p Value | Median survival (months) | |
|---|---|---|---|---|---|---|
| | | | | | Low | High |
| ACSL5 | OS | 1237 | 0.98 | 0.82 | 18.43 | 19.0 |
| | DFS | 278 | 1.29 | 0.046 | 11.23 | 10.03 |
| ANTXR1 | OS | 1237 | 1.01 | 0.928 | 18.57 | 18.97 |
| | DFS | 278 | 1.33 | 0.026 | 11.3 | 10.0 |
| AP1S3 | OS | 1237 | 1.2 | 0.008 | 20.03 | 16.43 |
| | DFS | 278 | 0.72 | 0.010 | 10.03 | 10.73 |
| ATP2C2 | OS | 1237 | 1.17 | 0.026 | 19.9 | 17.43 |
| | DFS | 278 | 0.82 | 0.126 | 10.13 | 10.43 |
| B3GNT5 | OS | 1237 | 1.4 | $2.10 \times 10^{-06}$ | 20.33 | 16.0 |
| | DFS | 278 | 0.89 | 0.377 | 10.6 | 10.13 |
| C15orf48 | OS | 1237 | 1.11 | 0.138 | 19.3 | 17.93 |
| | DFS | 278 | 1.31 | 0.033 | 11.47 | 9.23 |
| CAPG | OS | 1237 | 1.16 | 0.036 | 19.3 | 18.1 |
| | DFS | 278 | 0.66 | 0.001 | 9.77 | 11.4 |
| CTSK | OS | 1237 | 0.96 | 0.523 | 17.7 | 19.57 |
| | DFS | 278 | 1.38 | 0.012 | 11.4 | 10.03 |
| DAPP1 | OS | 1237 | 1.04 | 0.539 | 19.54 | 17.93 |
| | DFS | 278 | 1.63 | $1.98 \times 10^{-04}$ | 11.6 | 9.77 |
| DCBLD2 | OS | 1237 | 1.27 | $6.16 \times 10^{-04}$ | 20.9 | 16.0 |
| | DFS | 278 | 1.5 | 0.001 | 12.07 | 8.4 |
| GPX8 | OS | 1237 | 0.98 | 0.756 | 18.4 | 19.0 |
| | DFS | 278 | 1.64 | $1.40 \times 10^{-04}$ | 12.07 | 9.23 |
| HEPH | OS | 1237 | 1.03 | 0.705 | 18.97 | 18.0 |
| | DFS | 278 | 1.31 | 0.035 | 11.23 | 10.13 |
| IFI44 | OS | 1237 | 1.17 | 0.024 | 20.13 | 16.43 |
| | DFS | 278 | 1.16 | 0.251 | 11.23 | 9.83 |
| KRT23 | OS | 1237 | 1.23 | 0.003 | 19.93 | 17.6 |
| | DFS | 278 | 0.93 | 0.575 | 11.13 | 10.07 |
| NCF2 | OS | 1237 | 1.29 | $3.58 \times 10^{-04}$ | 19.73 | 17.0 |
| | DFS | 278 | 0.97 | 0.841 | 11.4 | 9.83 |
| OCIAD2 | OS | 1237 | 1.04 | 0.568 | 19.0 | 18.0 |
| | DFS | 278 | 1.65 | $9.86 \times 10^{-05}$ | 12.0 | 8.63 |
| SAMD9 | OS | 1237 | 1.09 | 0.201 | 19.8 | 17.27 |
| | DFS | 278 | 1.39 | 0.010 | 11.6 | 9.23 |
| SLC39A10 | OS | 1237 | 1.22 | 0.005 | 20.13 | 16.23 |
| | DFS | 278 | 1.01 | 0.947 | 10.5 | 10.13 |
| ST6GALNAC1 | OS | 1237 | 0.92 | 0.264 | 17.99 | 19.0 |
| | DFS | 278 | 0.77 | 0.036 | 10.03 | 11.23 |
| TBC1D2 | OS | 1237 | 1.16 | 0.034 | 20.0 | 16.87 |
| | DFS | 278 | 1.62 | $1.93 \times 10^{-04}$ | 11.73 | 9.13 |
| TMSB10 | OS | 1237 | 0.96 | 0.590 | 18.57 | 19.13 |
| | DFS | 278 | 1.36 | 0.015 | 11.6 | 9.23 |
| TSPAN5 | OS | 1237 | 1.51 | $6.30 \times 10^{-09}$ | 22.27 | 15.87 |
| | DFS | 278 | 1.17 | 0.209 | 11.3 | 9.1 |

*OS: overall survival, DFS: disease-free survival.

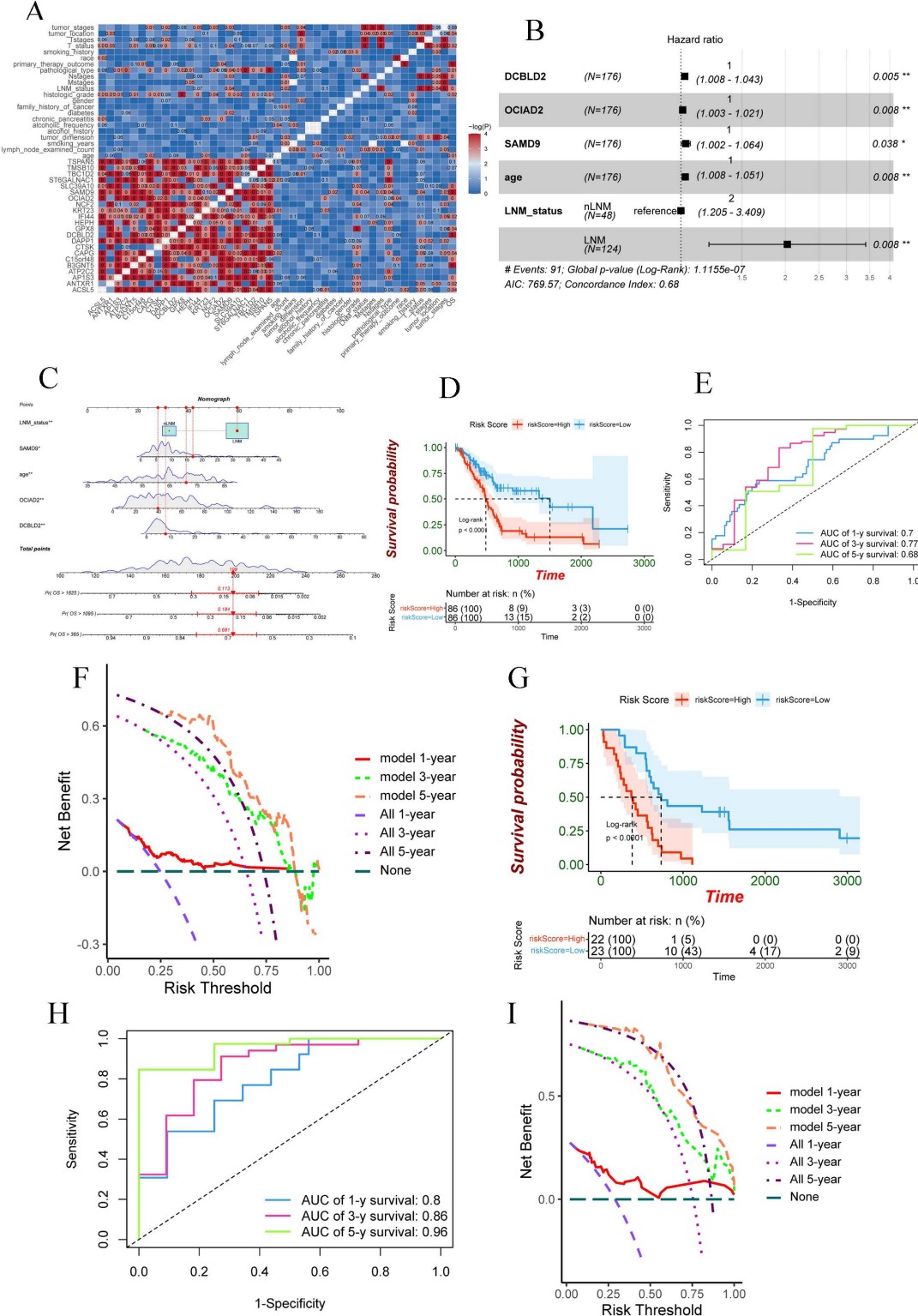

**Fig 2. *DCBLD2*, *OCIAD2*, and *SAMD9* are novel biomarkers with prognostic value.** (A) Correlation between the 22 DEGs and various clinicopathological features of pancreatic cancer patients. (B) Cox prognostic model based on *DCBLD2*, *OCIAD2*, *SAMD9*, age, and lymph node metastasis status. (C) Nomogram based on the Cox prognostic model to predict the 1,3, and 5-year survival rates of patients. (D) Patients with high prognostic model

scores had worse prognosis in TCGA pancreatic cancer dataset. (E) AUC of prognostic model predictive efficacy for 1,3, and 5-year survival in the TCGA pancreatic cancer dataset. (F) Clinical decision curve analysis results in the TCGA pancreatic cancer dataset. (G) Patients with high prognostic model scores in the GSE79668 dataset had a worse prognosis. (H) AUC of prognostic model predictive efficacy for 1,3, and 5-year survival in the GSE79668 dataset. (I) Results of clinical decision curve analysis in the GSE79668 dataset.

transcriptome sequencing data from 13 normal pancreatic tissues, 10 adjacent normal tissues, 57 primary tumors, and 13 metastatic tumors. A total of 225845 cells were obtained from these 93 tissues after quality control and filtration, and 29 cell clusters were obtained after clustering and grouping (Fig 3A). After annotation, it can be divided into 12 types of cells, including acinar cells, ductal cells, stellate cells, Schwann cells, fibroblasts, endothelial cells, T cells, B cells, macrophages, myeloid-derived suppressor cells (MDSCs), and mast cells (Fig 3A). Next, we analyzed the expression levels of *DCBLD2*, *OCIAD2,* and *SAMD9* in each tissue cell subpopulation. The results showed that *DCBLD2* mRNA expression was highest in ductal cells with metastatic cancer (Fig 3B and 3C). The expression of *OCIAD2* mRNA was highest in ductal cells and T cells of primary and metastatic tumors (Fig 3B and 3C). SAMD9 mRNA was highly expressed in both primary and metastatic ductal cells (Fig 3B and 3C), and was also higher in T cells from adjacent normal tissues (Fig 3B and 3C).

In addition, we found that, compared with DCBLD2 and SAMD9, OCIAD2 had the highest RNA and protein expression levels in PC tissues (Fig 4A and 4B). Therefore, we conducted further research on OCIAD2. For typical outcomes, BxPC-3 cell line with moderate OCIAD2 expression in pancreatic cancer cell lines was selected for subsequent experiments (Fig 4C).

## Knockdown of OCIAD2 in pancreatic cancer cells significantly inhibited JAK-STAT and cell cycle signaling pathways

To further understand the mechanism by which *OCIAD2* promotes PC progression, we performed transcriptomic sequencing analysis after knocking down *OCIAD2* expression in the BxPC-3 cell line. The results of transcriptome sequencing showed that the mRNA expression level of the *OCIAD2* knockdown group was significantly lower than in the control group in BxPC-3 cells (Fig 5A). PCA analysis of the transcriptome data showed that the first two principal components (PC1, PC2) could clearly distinguish different samples with *OCIAD2* knockdown from the control group (Fig 5B). To explore which pathways were inhibited or activated in the *OCIAD2* knockdown group compared to the control group, we used the run_wmean function of the decoupleR package [39] to analyze the transcriptome data of the two groups of samples. The results showed that the JAK-STAT signaling pathway was significantly inhibited in the *OCIAD2* knockdown group (Fig 5C). Next, we conducted differential gene expression analysis on the transcriptome data. The results showed that 466 genes were up-regulated and 522 genes were down-regulated in the *OCIAD2* knockdown group compared with the control group (Fig 5D and 5E). Based on DEGs and corresponding t values, we further analyzed the signaling pathways that were abnormally activated or inhibited between the *OCIAD2* knockdown group and control group using the run_mlm function. The results showed that the activities of JAK-STAT, PI3K, NFkB, and Androgen pathways were significantly inhibited in the *OCIAD2* knockdown group (Fig 5F). Then, we performed KEGG enrichment analysis of down-regulated DEGs in the *OCIAD2* knockdown group (Fig 5G and 5H). KEGG analysis showed that down-regulated DEGs were mainly enriched in cell cycle, oocyte meiosis, progesterone-mediated oocyte cellular senescence, and so on (Fig 5H). GSEA analysis was performed for all DEGs, and the results also showed that down-regulated genes were mainly enriched in the cell cycle pathway (Fig 5I and 5J).

## Knockdown of OCIAD2 significantly down-regulated STAT1 and STAT2 in JAK-STAT pathway and CCND1, CDK1 and CDK2 in cell cycle pathway

In order to further verify whether core genes in JAK-STAT and cell cycle signaling pathways are changed at the transcriptomic level, subsequent expression and correlation analysis were performed. Firstly, the expression levels of *JAK1*, *JAK2*,

PLOS Computational Biology

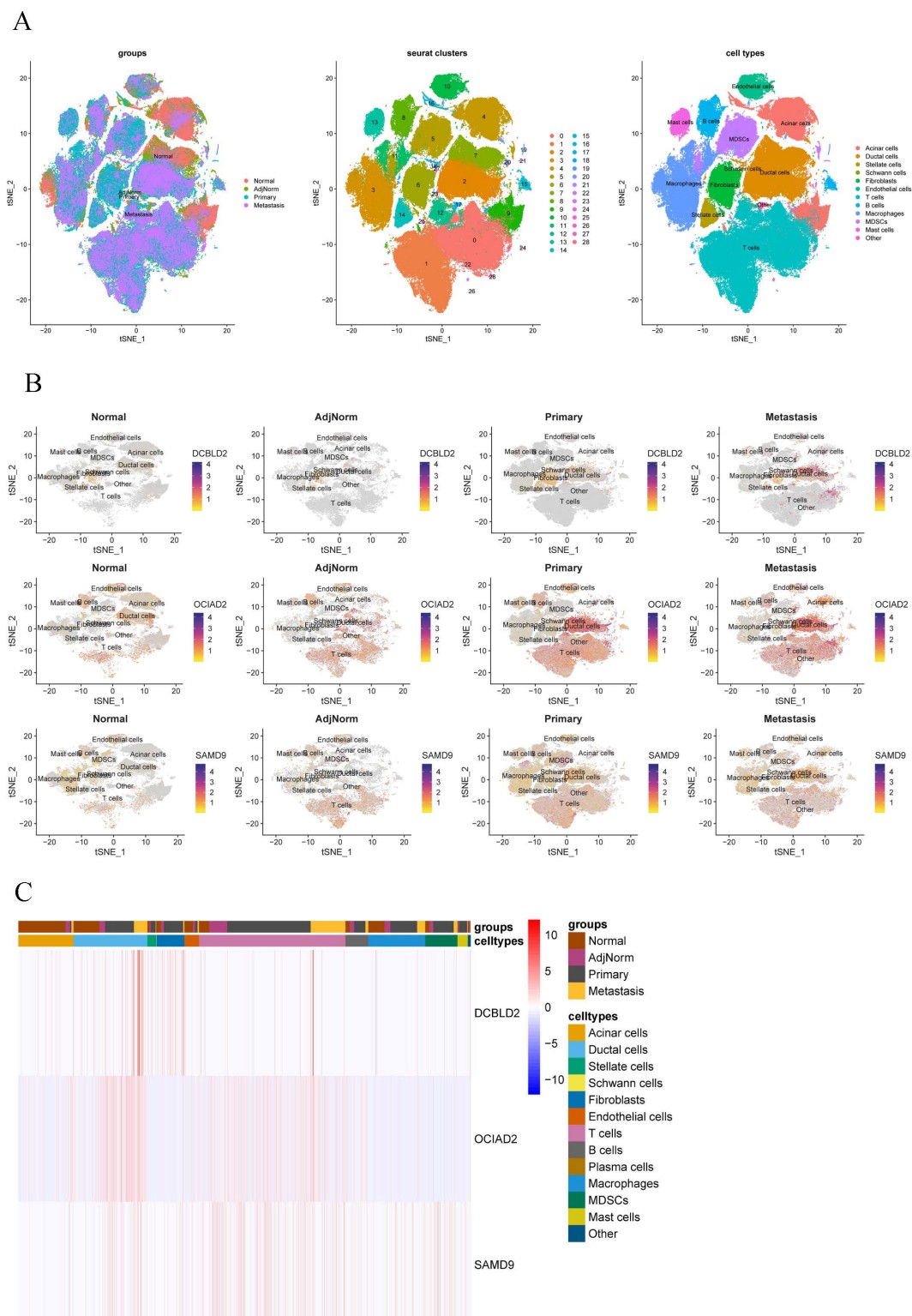

**Fig 3. The expression of *DCBLD2*, *OCIAD2* and *SAMD9* in pancreatic normal, adjacent normal, primary, and metastatic tumor tissues by single-cell sequencing.** (A) Dimensionality reduction cluster and cell subpopulation annotation results of 93 pancreatic normal, adjacent normal, primary, and metastatic tumor samples, combined with batch removal effect. (BC) After cell type annotation, the expression of *DCBLD2*, *OCIAD2*, and *SAMD9* was analyzed in normal pancreatic, adjacent normal, primary, and metastatic tumor tissues.

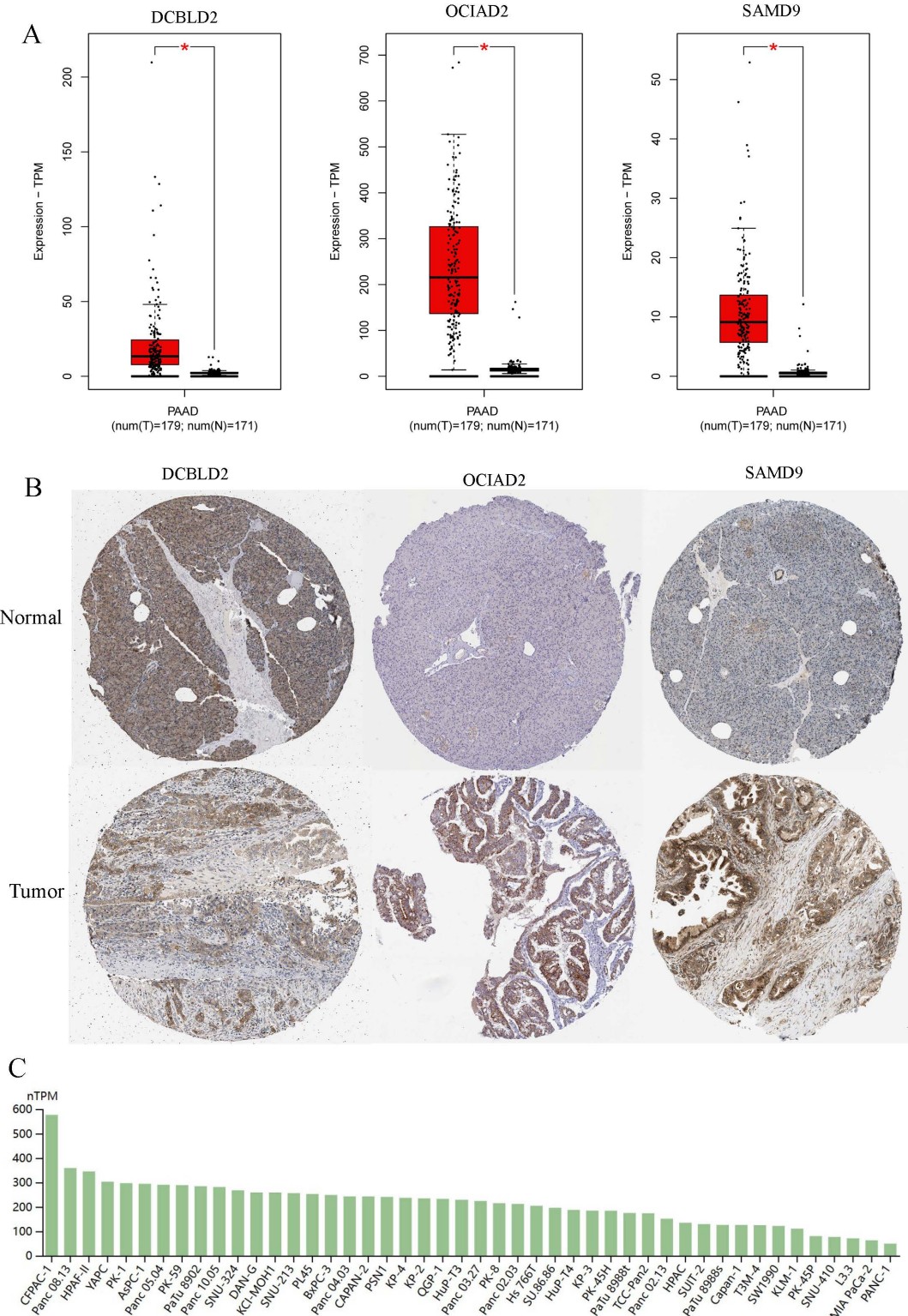

**Fig 4. The expression level of *DCBLD2, OCIAD2,* and *SAMD9* in pancreatic tumor tissue and cell lines.** (A) The mRNA expression levels of *DCBLD2*, *OCIAD2,* and *SAMD9* in pancreatic cancer and normal tissues (GEPIA). (B) The protein expression level of DCBLD2, OCIAD2, and SAMD9 in pancreatic cancer and normal tissues (HPA). (C) The mRNA expression levels of *OCIAD2* in 46 pancreatic cancer cell lines (HPA).

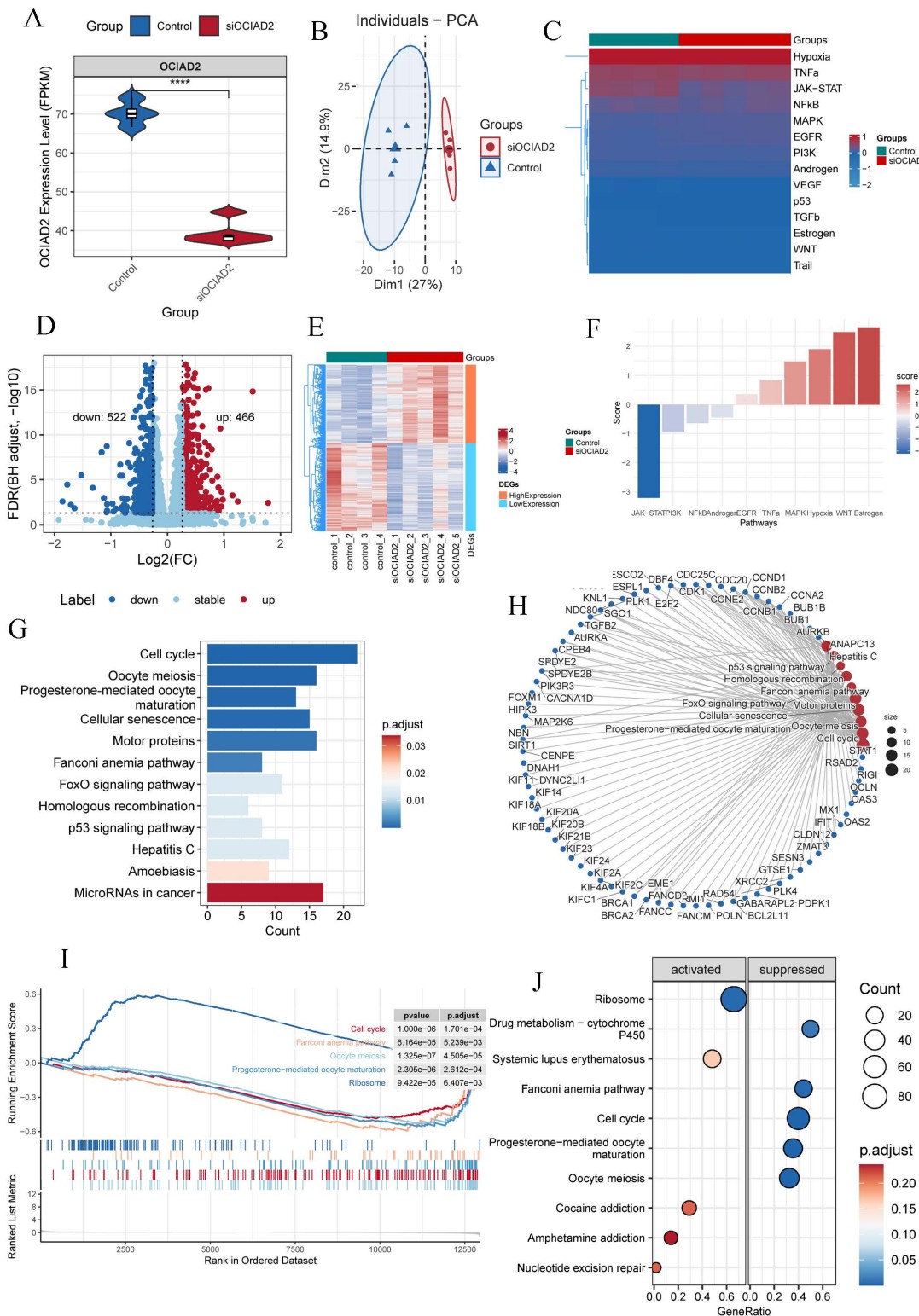

**Fig 5. The mechanism of targeting *OCIAD2* to inhibit the progression of pancreatic cancer.** (A) The expression level of *OCIAD2* mRNA in the siRNA interference group in the BxPC-3 cell line was significantly down-regulated compared with the control group. (B) Principal component analysis results at the transcriptome level of the two groups of samples. (C) The activation of 14 signaling pathways in the *OCIAD2* knockdown group. (D-E)

Volcano plot and heat map of up-and down-regulated DEGs in the *OCIAD2* knockdown group. (F) The activities of JAK-STAT, PI3K, NFkB, and Andro-gen pathways in the *OCIAD2* knockdown group. (G) KEGG enrichment analysis of down-regulated DEGs in the OCIAD2 knockdown group. (K) DEGs enriched in each major pathway in KEGG analysis. (I-J) GSEA analysis results of DEGs after *OCIAD2* knockdown.

*JAK3*, *TYK2*, *STAT1*, *STAT2*, *STAT3*, *STAT4*, *STAT5A*, *STAT5B* and *STAT6* in JAK-STAT pathway were verified after *OCIAD2* knockdown. The results revealed that both *STAT1* and *STAT2* were significantly down-regulated after knockdown of *OCIAD2*, whereas *JAK3*, *STAT5B,* and *STAT6* were all up-regulated (Fig 6A). Knockdown of *OCIAD2* significantly down-regulated *CCND1*, *CDK1,* and *CDK2* and up-regulated *CDK6* in the cell cycle signaling pathway (Fig 6B). Whether *OCIAD2* expression in real tumor tissues may also affect the expression of related genes in the JAK-STAT and cell cycle pathways is unknown, and we investigated this with data from GSE183795 [15]. The results showed that the expression levels of *STAT1*, *STAT2* and *STAT6* in JAK-STAT pathway were also significantly lower, while *STAT4*, *STAT5A* and *STAT5B* were higher in tumors with low *OCIAD2* expression (Fig 6C). Expression levels of *CCNA2*, *CCNB1*, *CCND1*, *CCNE1*, *CDK1*, *CDK2*, *CDK4,* and *CDK6* in the cell cycle pathway were all significantly lower in *OCIAD2* low-expressing tumor tissues, while *CDKN1B* was higher (Fig 6D). Based on these results, further analysis revealed a significant positive correlation between *OCIAD2* expression and the expression of *STAT1* and *STAT2* in the JAK-STAT pathway, and a significant negative correlation between *OCIAD2* expression and *STAT5B* (Fig 6E). The expression of *OCIAD2* was positively correlated with *CCND1*, *CDK1,* and *CDK2* in the cell cycle pathway (Fig 6F).

**The most sensitive compounds corresponding to different OCIAD2 expression levels**

To identify potential therapeutic agents for PC patients with different OCIAD2 expression, we performed drug sensitivity analyses (S2 Data). The analysis found that the top 10 drugs that were significantly positively correlated with *OCIAD2* expression were CIL70, CAY10618, decitabine, daporinad, SR1001, BIBR.1532, A.804598, StemRegenin.1, OSI.930, and gemcitabine (Fig 7A). It is suggested that these drugs may be more sensitive to patients with low expression of *OCIAD2*, but not suitable for patients with high expression of *OCIAD2* (Fig 7B). The top 10 drugs significantly negatively correlated with *OCIAD2* expression were MI.1, KHS101, GDC.0941, BRD.K27188169, pandacostat, ZSTK474, BRD.K80183349, tretinoin.carboplatin..2.1.mol.mol., navitoclax.pluripotin..1.1.mol.mol., and austocystin.D, respectively. (Fig 7C). This suggests that patients with high *OCIAD2* expression are more sensitive to these drugs, whereas patients with low expression are less sensitive (Fig 7D).

## Discussion

In this study, based on comprehensive transcriptomic and clinical data, we discovered many genes that are poorly understood but may be very important in pancreatic cancer. Meanwhile, at the transcriptome level, we revealed for the first time the association between *OCIAD2* and JAK-STAT1 as well as the cell cycle pathway. And the sensitivity of patients to various drugs under different *OCIAD2* expression levels was evaluated. Based on these results, we believe that *OCIAD2* is a potential prognostic and therapeutic marker for PC patients. Ovarian cancer immunoreactive antigen domain containing 2 (OCIAD2), with sequence similarity to *OCIAD1*, was first identified as a novel gene by the National Institutes of health mammalian gene collection program in 2002 [42]. *OCIAD2* is located on chromosome 4p11 in humans, with 7 exons and composed of 154 amino acids [43].

   *OCIAD2* has been reported to be implicated in liver cancer, lung adenocarcinoma, and ovarian mucinous tumors. Hypermethylation of *OCIAD2* in liver cancer tissues was associated with poor prognosis in patients [44–46]. In addition, Wu et al. showed that the hypermethylation of OCIAD2 in liver cancer resulted in the decrease of *OCIAD2* mRNA and protein levels, which promoted the migration and invasion of cancer cells, the enhancement of MMP9 expression, and the

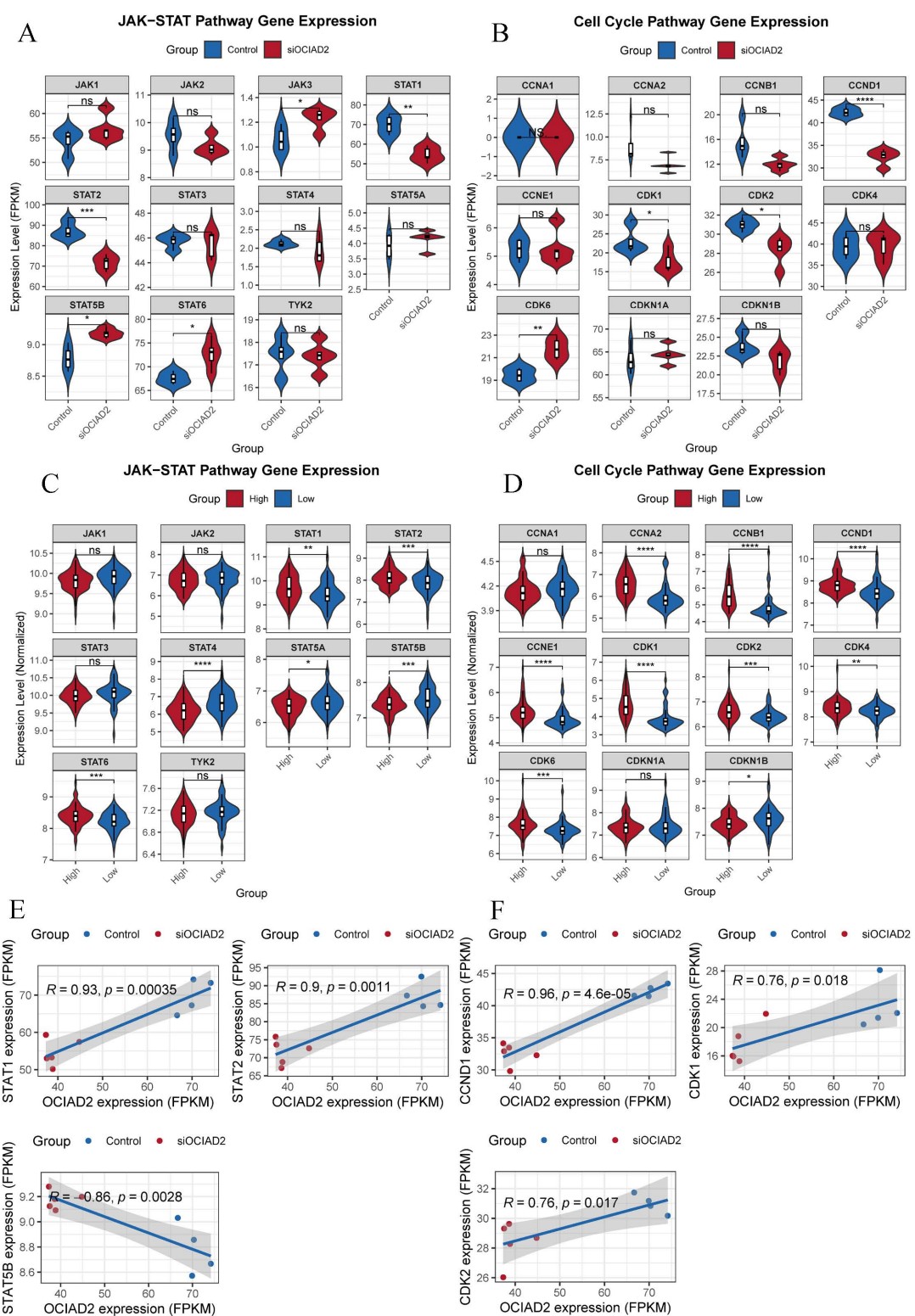

**Fig 6. Effects of *OCIAD2* on JAK-STAT and cell cycle signaling pathways.** (A, B) Alterations in core genes in the JAK-STAT and cell cycle pathways following knockdown of *OCIAD2* expression in the BxPC-3 cell line. (C-D) Expression levels of core genes in JAK-STAT and cell cycle pathways in PC tumor tissues in different *OCIAD2* groups. (E-F) Correlation of *OCIAD2* expression with *STAT1*, *STAT2*, *STAT5B*, *CCND1*, *CDK1* and *CDK2*.

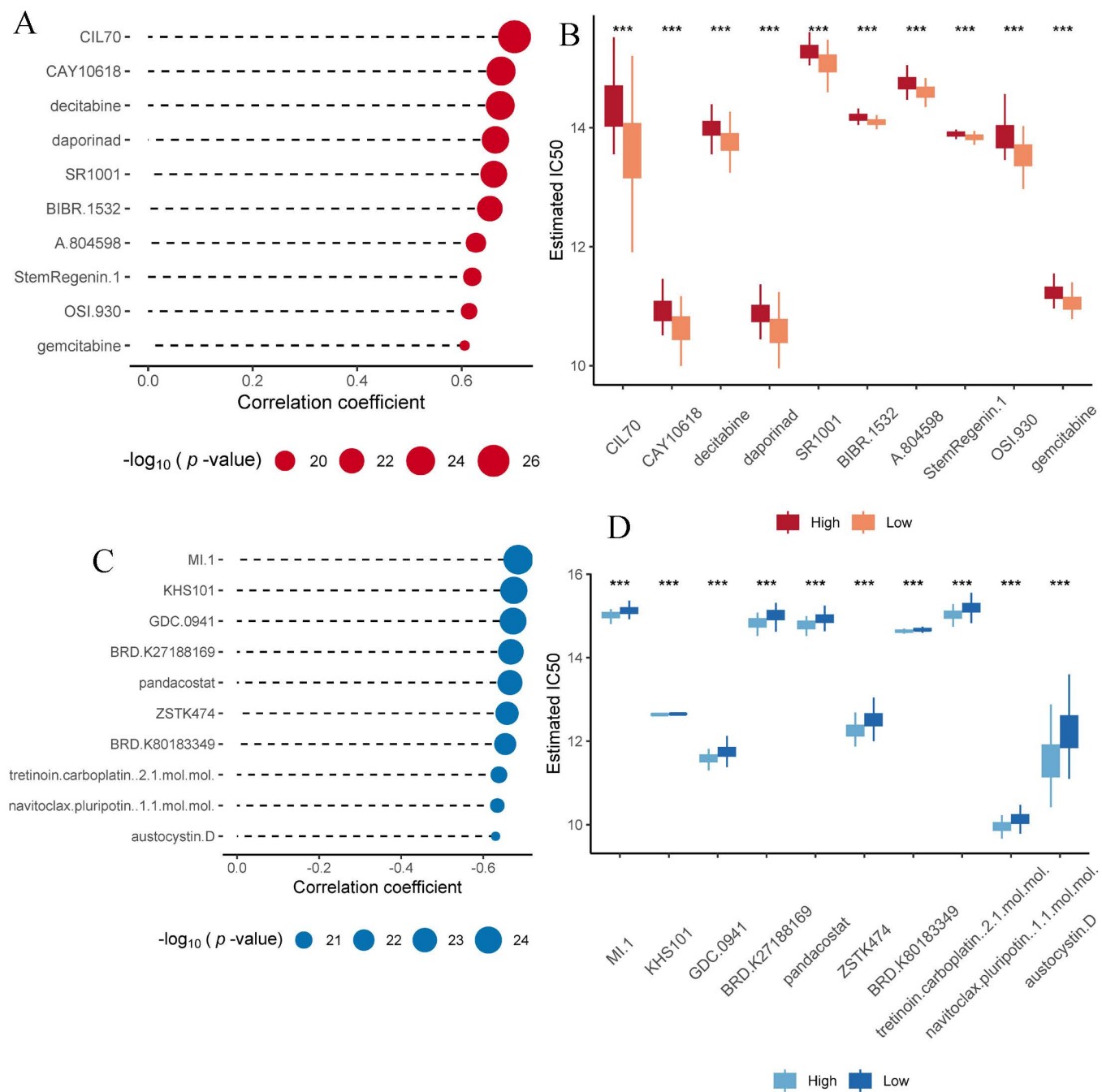

**Fig 7. The most sensitive compounds corresponding to PC patients with high and low *OCIAD2* expression.** (A) Top 10 compounds with significant positive correlations with *OCIAD2* expression. (B) IC50 values of positively correlated compounds at different *OCIAD2* levels. (C) Top 10 compounds with significant negative correlations with *OCIAD2* expression. (D) IC50 values of negatively correlated compounds at different *OCIAD2* levels.

activation of AKT and FAK [46]. Chigusa et al. demonstrated that similar to *OCIAD1*, *OCIAD2* was a cancer-associated protein whose expression increased during the progression of ovarian mucinous tumors and was a useful marker for evaluating malignancy [47].

Overexpression of *OCIAD2* was observed in lung adenocarcinoma, which was potentially caused by demethylation of the CpG site in the *OCIAD2* promoter region. Furthermore, low CpG methylation of *OCIAD2* was associated with adverse outcomes in patients [48]. Hong et al. reported that the expression of *OCIAD2* in invasive lung adenocarcinoma was significantly higher than in in situ lung adenocarcinoma and was associated with poor prognosis of patients. Inhibition of *OCIAD2* downregulated cell growth, proliferation, migration, and invasion, loss of mitochondrial structure, and reduction of mitochondrial number [49]. However, two studies on lung adenocarcinoma showed an inverse relationship between abnormal expression of *OCIAD2* and patient prognosis and clinicopathological features. One of them found that high *OCIAD2* protein expression was significantly correlated with vascular invasion, lymphatic infiltration, and pathological stages [50]. In another study, although OCIAD2 was highly expressed in lung adenocarcinoma, patients with high expression exhibited better prognosis. *OCIAD2* expression was inversely associated with lymphatic invasion, vascular invasion, and lymph node metastasis [51]. These results indicate that the function of *OCIAD2* in various tumors is complex, and its role varies among different tumors, requiring further research and exploration. In this study, we found that the expression level of *OCIAD2* was strongly correlated with chronic pancreatitis, primary therapy outcome, and pathological type in PC patients. In addition, we found that *OCIAD2* itself was rarely mutated, but PC patients with high *OCIAD2* expression had more mutations in *KRAS*, *TP53*, and *CDKN2A* than those with low *OCIAD2* expression (S2 Fig).

At the time we wrote the manuscript of this study, we found that Yi-Fan et al. had partially worked on *OCIAD2* in PC [52]. Although they also found that *OCIAD2* was highly expressed in PC and correlated with prognosis, they did not conduct a comprehensive evaluation of the prognostic value of *OCIAD2* in multiple datasets. In addition, they found that *OCIAD2* may play a role in PC cell proliferation, migration, and apoptosis through the PI3K/Akt signaling pathway. In this study, however, we found that JAK-STAT and the cell cycle pathway may play a more important role. In particular, we found that the JAK-STAT pathway is more strongly inhibited after *OCIAD2* knockdown than PI3K/Akt. In summary, these results suggest that *OCIAD2* plays a critical pathological function in PC and is a novel biomarker for the prognosis of PC patients.

However, our current study has several limitations. Although we demonstrated that *OCIAD2* plays an oncogenic role in PC and targeted knockdown of *OCIAD2* can inhibit the activity of JAK-STAT and the cell cycle, it has not been validated in vivo. In addition, although this study has confirmed the prognostic value of the mRNA level of *OCIAD2* for PC patients, the prognostic value of OCIAD2 protein expression in PC patients has not been investigated. These remain to be further studied in our future.

In conclusion, we have now demonstrated that *OCIAD2* is a useful prognostic biomarker for PC patients and plays a key pathological function in PC, and knockdown of *OCIAD2* significantly inhibits JAK-STAT and the cell cycle pathway activity. It represents a potential candidate for drug development in PC patients.

## Supporting information

**S1 Text. Table A.** Information of 7 GEO datasets for differential expression gene analysis. Table B. Significantly up-regulated and down-regulated DEGs in each GEO dataset.
(DOCX)

**S2 Text. The mRNA expression levels of *DCBLD2, OCIAD2*, and *SAMD9* in various tissues by single-cell sequencing.**
(DOCX)

**S1 Data. Differentially expressed genes analysis and intersection results of 7 geo datasets.**
(XLSX)

**S1 Fig. Confusion matrices of prognostic models 1, 3, and 5 years.**
(TIF)

**S2 Data. Correlation analysis results between *OCIAD2* expression and IC50 of different compounds in PC patients.**
(XLSX)

**S2 Fig. (A) The mutation rate of *OCIAD2* in patients with pancreatic cancer.** (B-C) Mutation profiles of PC patients with different *OCIAD2* expression levels.
(TIF)

**S3 Data. Table A.** Transcriptome sequencing results of knocking down *OCIAD2* expression in BxPC-3 cells. Table B. Analysis of differentially expressed genes after knocking down *OCIAD2* expression (DESeq2). Table C. Analysis of differentially expressed genes after knocking down *OCIAD2* expression (limma).
(XLSX)

## Acknowledgments

We are grateful to the generous data contributors and the developers and maintainers of the public databases and R packages.

## Author contributions

**Conceptualization:** Zhongyuan Cui, Zhixian Wu, Xiaojun Huang.

**Data curation:** Zhongyuan Cui.

**Formal analysis:** Zhongyuan Cui, Xia Lei.

**Funding acquisition:** Yani Gou, Zhixian Wu, Xiaojun Huang.

**Investigation:** Zhongyuan Cui, Xia Lei, Yani Gou.

**Methodology:** Zhongyuan Cui, Xia Lei, Zhixian Wu.

**Project administration:** Xiaojun Huang.

**Resources:** Yani Gou, Zhixian Wu, Xiaojun Huang.

**Supervision:** Xiaojun Huang.

**Visualization:** Zhongyuan Cui.

**Writing – original draft:** Zhongyuan Cui.

**Writing – review & editing:** Zhixian Wu, Xiaojun Huang.

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
