## [Decision Letter · Decision Letter 0]

24 Jul 2025

PCOMPBIOL-D-25-01246

OCIAD2 as a novel prognostic and therapeutic biomarker for pancreatic cancer: a study based on transcriptomic signature and bioinformatics analysis

PLOS Computational Biology

Dear Dr. Cui,

Thank you for submitting your manuscript to PLOS Computational Biology. After careful consideration, we feel that it has merit but does not fully meet PLOS Computational Biology's publication criteria as it currently stands. Therefore, we invite you to submit a revised version of the manuscript that addresses the points raised during the review process.

Please submit your revised manuscript within 60 days Sep 23 2025 11:59PM. If you will need more time than this to complete your revisions, please reply to this message or contact the journal office at ploscompbiol@plos.org. Please include the following items when submitting your revised manuscript:

We look forward to receiving your revised manuscript.

Kind regards,

Hao Hu, Ph.D

Academic Editor

PLOS Computational Biology

Ilya Ioshikhes

Section Editor

PLOS Computational Biology

**Journal Requirements:**

At this stage, the following Authors/Authors require contributions: Zhongyuan Cui, Xia Lei, Yani Gou, Xiaojun Huang, and Zhixian Wu. Please ensure that the full contributions of each author are acknowledged in the "Add/Edit/Remove Authors" section of our submission form.

4) Your current Financial Disclosure states, "The author(s) received no specific funding for this work."

However, your funding information on the submission form indicates receiving funds.

Please ensure that the funders and grant numbers match between the Financial Disclosure field and the Funding Information tab in your submission form. Note that the funders must be provided in the same order in both places as well. 

1) Please clarify all sources of financial support for your study. List the grants, grant numbers, and organizations that funded your study, including funding received from your institution. Please note that suppliers of material support, including research materials, should be recognized in the Acknowledgements section rather than in the Financial Disclosure

2) State the initials, alongside each funding source, of each author to receive each grant. For example: "This work was supported by the National Institutes of Health (####### to AM; ###### to CJ) and the National Science Foundation (###### to AM)."

3) State what role the funders took in the study. If the funders had no role in your study, please state: "The funders had no role in study design, data collection and analysis, decision to publish, or preparation of the manuscript."

4) If any authors received a salary from any of your funders, please state which authors and which funders..

**Reviewers' comments:**

Reviewer's Responses to Questions

Reviewer #1: In this article, extensive research was conducted on pancreatic cancer through a large amount of transcriptome data. The results revealed many genes that are poorly understood but potentially important in pancreatic can cer. Further studies in the BxPC3 cell line based on high-throughput sequencing revealed a relationship between OCIAD2 and the over-activation of STAT and cell cycle pathways. Furthermore, its conclusion has been verified in the transcriptome and clinical data of pancreatic cancer in other independent studies. These findings have good clinical significance and deserve attention. However, the following issues still need to be further addressed:

1、The mutation of OCIAD2 in pancreatic cancer can be analyzed at the genomic level. For example, the mutation of OCIAD2 itself in PC, and the characteristics of the mutation spectrum in patients with different OCIAD2 expression levels.

2、Gene set enrichment analysis should be abbreviated as GESA when it first appears in the method

3、Figure 5J shows incomplete and needs to be resized.

4、Some expressions in the method are not standardized. For example, the writing of "siOCIAD2" in the method RNA sequencing and bioinformatics analysis is unclear and needs to be corrected.

5、Given the complex role of OCIAD2 in tumors such as lung cancer, the discussion of the relationship between OCIAD2 and clinicopathological features of PC patients should be increased.

6、It is suggested to further refine the language of the article.

Reviewer #2: The study of "OCIAD2 as a novel prognostic and therapeutic biomarker for pancreatic cancer: a study based on transcriptomic signature and bioinformatics analysis" combined multiple datasets of pancreatic cancer and identified OCIAD2 as a poential biomarker for prognosis and therapy. However, the value of this study is still limited.

1. There have already been too many similar reports for pancreatic cancer, which severely challenged the novelty of this study. Just name a few (still many others):

1) JagadeeswaraRao 2024 An integrated ensemble learning technique for gene expression classification and biomarker identification from RNA-seq data for pancreatic cancer prognosis, International Journal of Information Technology

2) Hossen 2023 Robust identification of common genomic biomarkers from multiple gene expression profiles for the prognosis, diagnosis, and therapies of pancreatic cancer, Computers in Biology and Medicine

3) Zhuang 2021 Identification of LIPH as an unfavorable biomarkers correlated with immune suppression or evasion in pancreatic cancer based on RNA-seq, Cancer Immunology, Immunotherapy

4) Xu 2021 Development and clinical validation of a novel 9-gene prognostic model based on multi-omics in pancreatic adenocarcinoma, Pharmacological Research

5) Atay 2020 Integrated transcriptome meta-analysis of pancreatic ductal adenocarcinoma and matched adjacent pancreatic tissues, PeerJ

6) Zhang 2020 Early Diagnosis of Pancreatic Ductal Adenocarcinoma by Combining Relative Expression Orderings With Machine-Learning Method, Frontiers in Cell and Developmental Biology

7) Cheng 2019 Identification of candidate diagnostic and prognostic biomarkers for pancreatic carcinoma, eBioMedicine

2. More seriously, the Yin 2024 Gene paper (ref. 52), reported very similar discoveries on OCIAD2, although some differences in the proposed downstream mechanisms with the authors.

3. Even for the mechanisms, the data are very weak. It is unclear how OCIAD2 regulates the hypothesized JAK-STAT pathway in PCa cell lines or specimens.

4. The therapeutic value of OCIAD2 is also limited. No specific drugs for patients with OCIAD2 overexpression or silencing were validated.

Reviewer #3: The authors used OCIAD2 as a novel prognostic and therapeutic biomarker for pancreatic cancer and conducted a

study based on transcriptomic signature and bioinformatics analysis. This work is interesting, and the paper need to be revised as follows:

1 To strengthen the findings, the authors should also consider Friedman’s test and Shapiro–Wilk test

statistical tests.

2 Provide a confusion matrix of the proposed model atleast. This would help

illustrate the distribution of true positives, false positives, true negatives, and

false negatives, in a clearer view of model performance across different

classes

3 The experimental results could further be clearly presented, either as an additional column in the table or in a separate statistical comparison section

**Have the authors made all data and (if applicable) computational code underlying the findings in their manuscript fully available?**

Reviewer #1: Yes

Reviewer #2: Yes

Reviewer #3: None

PLOS authors have the option to publish the peer review history of their article (what does this mean? ). If published, this will include your full peer review and any attached files.

**Do you want your identity to be public for this peer review?** For information about this choice, including consent withdrawal, please see our Privacy Policy .

Reviewer #1: No

Reviewer #2: No

Reviewer #3: No

**Figure resubmission:**

**Reproducibility:**



---

## [Decision Letter · Decision Letter 1]

4 Sep 2025

PCOMPBIOL-D-25-01246R1

OCIAD2 as a novel prognostic and therapeutic biomarker for pancreatic cancer: a study based on transcriptomic signature and bioinformatics analysis

PLOS Computational Biology

Dear Dr. Cui,

Thank you for submitting your manuscript to PLOS Computational Biology. After careful consideration, we feel that it has merit but does not fully meet PLOS Computational Biology's publication criteria as it currently stands. Therefore, we invite you to submit a revised version of the manuscript that addresses the points raised during the review process.

Please submit your revised manuscript within 60 days Nov 04 2025 11:59PM. If you will need more time than this to complete your revisions, please reply to this message or contact the journal office at ploscompbiol@plos.org. Please include the following items when submitting your revised manuscript:

We look forward to receiving your revised manuscript.

Kind regards,

Hao Hu, Ph.D

Academic Editor

PLOS Computational Biology

Ilya Ioshikhes

Section Editor

PLOS Computational Biology

**Journal Requirements:**

Please provide an Author Summary. This should appear in your manuscript between the Abstract (if applicable) and the Introduction, and should be 150-200 words long. The aim should be to make your findings accessible to a wide audience that includes both scientists and non-scientists. Sample summaries can be found on our website under Submission Guidelines:

**Reviewers' comments:**

Reviewer's Responses to Questions

**Comments to the Authors:**

Reviewer #1: The authors have addressed my concerns.

Reviewer #2: As I have mentioned in the previous comments, the current study needs to demonstrate its novelty against a number of similar publications. The mechanism part, which could be a good direction, remains weak in this revised version. Actually, it has been reported that OCIAD2 directly interacts with and activates STAT3 (https://pmc.ncbi.nlm.nih.gov/articles/PMC5943604/). I suggest the authors to verify whether this mechanism is the one in PAAD as well.

Reviewer #3: The author has carefully revised the paper according to the comments and suggestions

**Have the authors made all data and (if applicable) computational code underlying the findings in their manuscript fully available?**

Reviewer #1: Yes

Reviewer #2: Yes

Reviewer #3: None

PLOS authors have the option to publish the peer review history of their article (what does this mean? ). If published, this will include your full peer review and any attached files.

**Do you want your identity to be public for this peer review?** For information about this choice, including consent withdrawal, please see our Privacy Policy .

Reviewer #1: **Yes: ** Hongwei Cheng

Reviewer #2: No

Reviewer #3: No

**Figure resubmission:**

**Reproducibility:**



---

## [Decision Letter · Decision Letter 2]

28 Sep 2025

Dear Dr. Cui,

We are pleased to inform you that your manuscript 'OCIAD2 as a novel prognostic and therapeutic biomarker for pancreatic cancer: a study based on transcriptomic signature and bioinformatics analysis' has been provisionally accepted for publication in PLOS Computational Biology.

Best regards,

Hao Hu, Ph.D

Academic Editor

PLOS Computational Biology

Ilya Ioshikhes

Section Editor

PLOS Computational Biology

Reviewer's Responses to Questions

**Comments to the Authors:**

Reviewer #2: Thanks for addressing my concerns.

**Have the authors made all data and (if applicable) computational code underlying the findings in their manuscript fully available?**

Reviewer #2: Yes

PLOS authors have the option to publish the peer review history of their article (what does this mean? ). If published, this will include your full peer review and any attached files.

**Do you want your identity to be public for this peer review?** For information about this choice, including consent withdrawal, please see our Privacy Policy .

Reviewer #2: No

---

## [Editor Report · Acceptance letter]

PCOMPBIOL-D-25-01246R2

OCIAD2 as a novel prognostic and therapeutic biomarker for pancreatic cancer: a study based on transcriptomic signature and bioinformatics analysis

Dear Dr Cui,

I am pleased to inform you that your manuscript has been formally accepted for publication in PLOS Computational Biology. Your manuscript is now with our production department and you will be notified of the publication date in due course.

With kind regards,

Zsofia Freund
